# Comparison of Post-Vaccination Response (Humoral and Cellular) to BNT162b2 in Clinical Cases, Kidney and Pancreas Transplant Recipient with Immunocompetent Subjects over Almost Two Years of Parallel Monitoring

**DOI:** 10.3390/vaccines12080844

**Published:** 2024-07-26

**Authors:** Jaroslaw Walory, Iza Ksiazek, Karolina Wegrzynska, Anna Baraniak

**Affiliations:** 1Department of Pharmaceutical Microbiology and Laboratory Diagnostics, National Medicines Institute, 00-725 Warsaw, Poland; k.wegrzynska@nil.gov.pl; 2Department of Biochemistry and Biopharmaceuticals, National Medicines Institute, 00-725 Warsaw, Poland; i.ksiazek@nil.gov.pl

**Keywords:** anti-SARS-CoV-2 antibodies, cellular response, humoral response, immunocompromised patient, immunosuppression, IFN γ, mRNA vaccine

## Abstract

Background: Vaccination is one of the most effective medical interventions to prevent infectious diseases. The introduction of vaccines against coronavirus acute respiratory syndrome 2 (SARS-CoV-2) was aimed at preventing severe illness and death due to coronavirus disease 2019 (COVID-19). Solid organ transplant recipients (SOTRs) are at high risk of infection with SARS-CoV-2 and serious effects associated with COVID-19, mainly due to the use of immunosuppressive therapies, which further cause suboptimal response to COVID-19 vaccination. Aim of the study: We aimed to compare post-vaccination response to BNT162b2 in kidney–pancreas transplant recipient, specifically in immunocompetent individuals, over two years of simultaneous monitoring. Methods: To determine the humoral response, the levels of the IgG and IgA anti-S1 antibodies were measured. To assess the cellular response to SARS-CoV-2, the released IFN-γ-S1 was determinate. Results and Conclusion: After primary vaccination, compared to immunocompetent subjects, SOTR showed lower seroconversion for both antibody classes. Only the additional dose produced antibodies at the level reached by the control group after the baseline vaccination. During the monitored period, SOTR did not achieve a positive cellular response in contrast to immunocompetent individuals, so in order to obtain longer protection, including immune memory, the adoption of booster doses of the vaccine should be considered.

## 1. Introduction

Coronavirus disease 2019 (COVID-19), caused by severe acute respiratory syndrome coronavirus 2 (SARS-CoV-2), has affected hundreds of millions of people and significantly increased mortality worldwide [1]. Solid organ transplant recipients (SOTRs) belong to a group of patients at higher risk of SARS-CoV-2 infection and severe COVID-19-related outcomes, including death, mainly due to immunosuppressive treatment to protect organ rejection [2,3,4]. In addition, receiving immunosuppressants is usually associated with a suboptimal response to vaccination and this, combined with a systematic decline in SARS-CoV-2 antibody levels over time, means that although COVID-19 vaccines reduced the risk of morbidity and mortality in SOTRs, the rates of breakthrough infection and death remained higher in this population compared to immunocompetent individuals [5,6,7]. All of this makes the monitoring of the post-vaccination response in this group of patients particularly important and serves to identify poor or non-responders to vaccination and to assess the need for booster doses to enhance and prolong the duration of the specific immune response.

Here, we present a comprehensive characterization of the post-vaccination response (both humoral and cellular) to BNT162b2 (Comirnaty^®^, Pfizer, Philadelphia, PA, USA and BioNTech, Mainz, Germany) in a kidney–pancreas transplant recipient. Monitoring of the immune response was carried out for 20 months (seven checkpoints in total), and the individual results were compared with the mean of measurements simultaneously performed in immunocompetent subjects for whom partial results (antibody levels) have been previously published [8].

## 2. Materials and Methods

### 2.1. Study Design, Data Collection, and Participant Characteristics

The investigation was designed to compare the post-vaccination response (both humoral and cellular) to BNT162b2 in kidney–pancreas transplant recipients with immunocompetent individuals over nearly two years of simultaneous monitoring. The timeline of vaccine administrations and serum collections for the immune response studies, for both the SOTR case and the control group, is shown in Figure 1.

### 2.2. SARS-CoV-2 Testing of SOTR Case

Due to cases of SARS-CoV-2 infection among the closest co-workers, the SOTR case was screened for SARS-CoV-2 five times (once in 2020, three times in 2022 and once in 2023) with RT-qPCRs using MutaPLEX^®^ Coronavirus (SARS-CoV-2) kit (Immundiagnostik AG, Bensheim, Germany), as described previously [9,10]. The positive sample was tested for variant diversity using the Bosphore^®^ SARS-CoV-2 Variant Detection Kit v1 assay (Anatolia Geneworks, Istanbul, Turkey), ID ™ SARS-CoV-2/VOC evolution Pentaplex kit (ID Solutions, Grabels, France) and PKamp VariantDetect SARS-CoV-2 RT-PCR KIT Combination 6 (PerkinElmer, Waltham, MA, USA), as we reported earlier [11]. Additionally, the SOTR case tested itself twice with commercial rapid antigen tests (RATs) in 2023.

### 2.3. Anti-SARS-CoV-2 Immune Response Tests

Both humoral and cellular responses were investigated by enzyme-linked immunosorbent assays (ELISA) with an Infinite^®^ M1000 PRO (Tecan Trading AG, Männedor, Switzerland). To determine the humoral response, IgA and IgG antibodies against spike (S) protein (IgA-S1 and IgG-S1) and IgG antibodies to nucleocapsid (N) protein (IgG-NCP) were tested seven times (except for IgA-S1, for which measurements were abandoned at the last checkpoint due to their low levels) using three commercial assays: Anti-SARS-CoV-2 QuantiVac-ELISA (IgG), Anti-SARS-CoV-2 ELISA (IgA), and Anti-SARS-CoV-2-NCP ELISA (IgG) (EUROIMMUN Medizinische Labordiagnostika AG, Lübeck, Germany), as described earlier [8,10]. The immune cellular response was assessed using the Quan-T-Cell SARS-CoV-2 assay (IGRA, Interferon Gamma Release Assay, EUROIMMUN Medizinische Labordiagnostika AG, Lübeck, Germany). Overall cellular response activity was established by measuring gamma interferon (IFN-γ) released by T lymphocytes after non-specific mitogen stimulation. In contrast, to determine the specific cellular response to SARS-CoV-2, T-cells were treated with S1 antigen of SARS-CoV-2 and the released IFN-γ-S1 was measured. The tests were performed and interpreted according to the manufacturer’s instructions.

## 3. Results

### 3.1. Participant Characteristics

A total of 92 participants were included in the study: 1 SOTR case and 91 immunocompetent individuals. The number of study subjects is representative of the Polish adult population (30 million), with a confidence level of α = 0.95 and a maximum error in the results of up to 10%. The participants in the control group were characterized in detail in a previous study [8] and consisted of 72 females (79%) and 19 males (21%) with a median age of 49 years. The SOTR case was represented by a 41-year-old female healthcare administrative worker with a seven-year status as a recipient of a simultaneous kidney–pancreas transplant due to type 1 diabetes mellitus complicated by diabetic nephropathy, on a chronic immunosuppressive regimen. Maintenance immunosuppression consisted of tacrolimus (3 mg/day), mycophenolate mofetil (MMF; 1 g/day), and prednisolone (2.5 mg/day). Furthermore, the woman had hypertension treated with bisoprolol. She was the optimal weight, with a body mass index of 24.17. Details of the SOTR’s basic medical history are shown in Figure 2.

### 3.2. SARS-CoV-2 Detection and Variant Identification

Only one of the performed SARS-CoV-2 RT-qPCR tests was positive (21 March 2022) and the Omicron variant was identified. At that time, the SOTR complained of losing their sense of taste for three days. Additionally, one RAT test yielded a positive result (29 March 2023). The patient passed the infection asymptomatically.

### 3.3. Anti-SARS-CoV-2 Immune Response

The levels of IgG-S1 and IgA-S1 antibodies obtained for the SOTR and previously published averaged results of the control group [8] are shown in Table 1.

The results of IgG-S1 and IgA-S1 antibody levels indicated that the SOTR had no previous SARS-CoV-2 infection until the start of the study (recovered patients showed several times higher antibody levels in both classes [8]). After primary vaccination, its antibody levels increased only slightly: 2.7- and 16-fold for IgG-S1 and IgA-S1, respectively. Interestingly, IgG class antibody levels further doubled at the third checkpoint and remained in a similar range until the fourth measurement. In contrast, IgA class antibody levels declined steadily at analogous control points. Importantly, compared to the control group at the second checkpoint, the SOTR case demonstrated almost 100- and 5-fold lower seroconversions for IgG-S1 and IgA-S1, respectively. At the fifth serum collection, just after the woman received the additional dose, more than 250- and 100-fold increases (compared to the baseline values from the first measurements) were found in the antibody levels for IgG-S1 and IgA-S1, respectively. Four months after the additional dose (sixth serum collection), she showed a more than 2.6-fold decrease in antibody levels in both classes. In contrast, at the seventh checkpoint, an almost two-fold increase (compared to the earlier measurement) in IgG-S1 level was observed, most likely as a result of SARS-CoV-2 infection in March 2022. Interestingly, throughout the woman’s monitoring (20 months, seven checkpoints), all her IgG-NCP results were negative. The dynamics of the antibody response are shown in Figure 3A–C.

The cellular responses were determined three times, in October 2021, February 2022 and October 2022 (at the fifth, sixth, and seventh checkpoints, respectively). The levels of IFNγ-S1 and non-specific IFNγ obtained for the SOTR case and control group are shown in Table 2.

All cellular response results of the woman were negative. The initial IFNγ-S1 level obtained for her was 82.5 times lower than the average titer for the control group. At subsequent checkpoints, the determined levels increased slightly, but they were always several times lower than for the control group (40- and 87-fold for the sixth and seventh checkpoints, respectively). In addition, SOTR’s overall cellular immune response was about 10 times lower in each measurement than in the control group. The kinetics of normalized IFNγ-S1 data are shown in Figure 3D.

## 4. Discussion

Since SOTRs are characterized by a poor response to many subunit protein vaccines, the woman’s suboptimal response to BNT162b2 primary vaccination was not entirely unexpected [5,12]. Individuals undergoing maintenance immunosuppression are usually prescribed a calcineurin inhibitor, which interferes with the activation of T cells, and an anti-metabolite that non-specifically blocks the proliferation of activated T and B cells [12]. Our patient received drugs from both groups, tacrolimus and MMF, respectively. In addition, the woman was also treated with prednisolone, a corticosteroid that may also suppress the immune response. Asderakis et al. showed that the presence of prednisolone inhibited antibody development among patients who received the BNT162b2, while it had no effect on the seroconversion rate among those who received another mRNA vaccine. Furthermore, subjects receiving both MMF and prednisolone had the lowest humoral response among the tested SOTRs [5].

The seroconversion rate in the study SOTR case after two vaccine doses was very low (the levels of IgG and IgA antibodies were only 1% and 20% of the titers obtained in the control group, respectively) and the woman was classified as a vaccine non-responder. Poor or no humoral response to primary vaccination in SOTRs was also reported by other researchers [7,13,14]. Importantly, just a few days after receiving the additional dose, the woman experienced an increase in antibody levels of both classes and they were 101% and 144% of the IgG and IgA levels obtained for the control group after primary vaccination, respectively. Our observations regarding the enhancement of the immune response by the additional dose are consistent with other reports [7,15,16]. However, it should be noted that multiple vaccinations do not always result in an increase in antibody levels in SOTRs [17,18]. Overall, seroconversion rates after COVID-19 vaccination in this group of patients vary from reports and mainly depend on individual characteristics, the type of transplanted organ, the medications taken, the kind of vaccine used, and the number of administered doses [5,7,17,19,20]. Four months after the rise in antibody levels following the additional dose, the woman’s antibody levels began to decline, and at the sixth checkpoint they were twice as low as they were at the previous one. This may have contributed to the breakthrough infection identified in her during this time, but importantly, she passed it very gently. The breakthrough infection was caused by the Omicron variant, and studies evaluating the humoral response of SOTRs against SARS-CoV-2 variants showed that additional doses of vaccination significantly improve the antibody response against all variants except Omicron [21,22,23,24]. As expected, the woman’s antibody levels increased after infection. In contrast to the humoral response, the woman did not develop a cellular response. Although her IFNγ-S1 levels were very low, they increased slightly with each checkpoint, which may have been predictive of a positive SARS-CoV-2-specific T-cell response over time. Unfortunately, the SOTR case did not receive booster doses, which are currently recommended by the European Centre for Disease Prevention and Control (ECDC) in the COVID-19 vaccination schedule for immunocompromised individuals [25]. As with humoral response, cellular response studies in this group of patients vary from report to report and depend on the previously mentioned factors [7,16].

A limitation of our study was that our assays were conducted on only one SOTR case. On the other hand, an important aspect of this research was that we were able to compare the post-vaccination immune response, both humoral and cellular, in a clinical case of a kidney–pancreas transplant recipient with immunocompetent individuals, since the study was conducted at the same time. In addition, we presented specific cellular response data normalized to the patient’s total cellular response, allowing comparison with results obtained by other researchers employing other assays.

## 5. Conclusions

Although our SOTR patient did not develop a humoral response after primary BNT162b2 vaccination, the additional dose caused the woman’s antibody levels to reach those achieved in immunocompetent individuals after primary doses. During the monitored period, unlike the humoral response, the woman did not achieve a positive cellular response, so to obtain longer protection with the involvement of immunological memory, it is worth considering booster doses of the vaccine, which are recommended by the ECDC in the COVID-19 vaccination schedule for immunocompromised patients.

## Figures and Tables

**Figure 1 vaccines-12-00844-f001:**
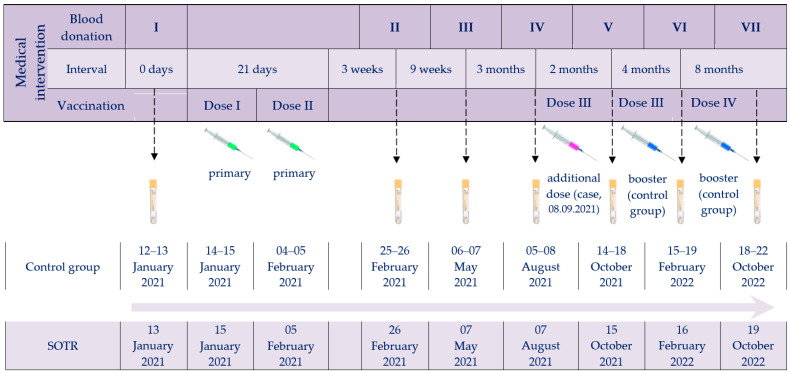
Timeline of vaccine administrations and serum collection checkpoints. Abbreviations: Roman numerals (I–VII), arrows and tubes, subsequent blood donations; green syringes, primary vaccination; purple syringe, SOTR’s additional dose; blue syringes, booster doses of the control group.

**Figure 2 vaccines-12-00844-f002:**
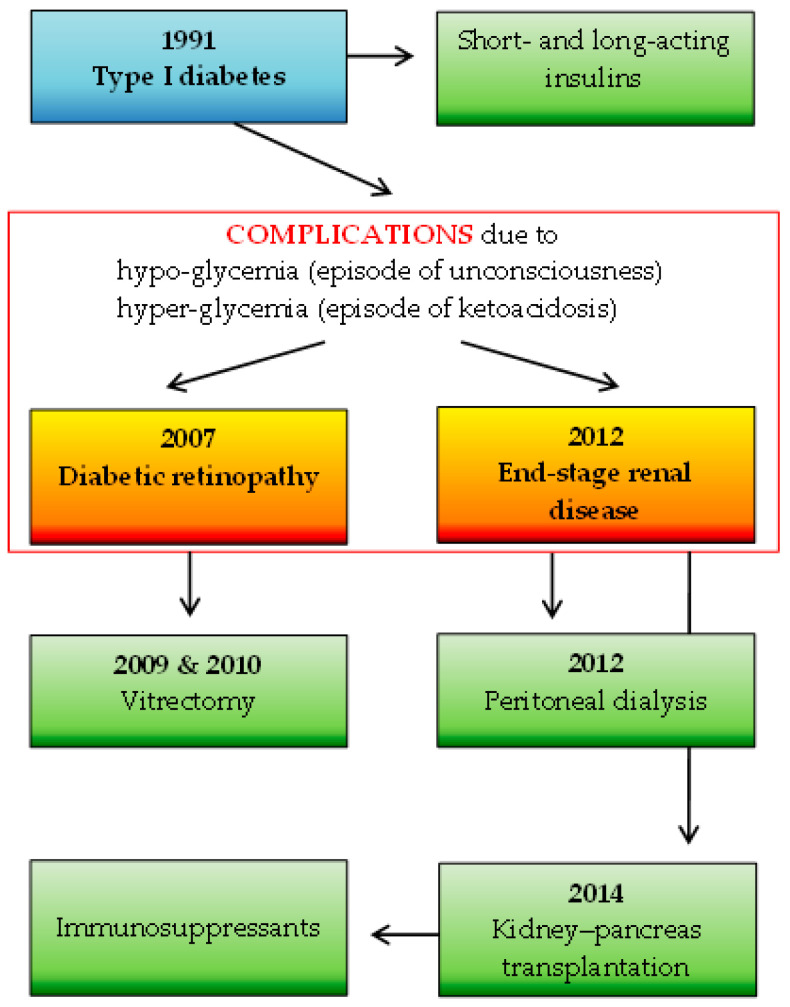
The SOTR’s medical history: the basic disease (marked in blue) along with complications (marked in red frame) and applied specific therapies (marked in green).

**Figure 3 vaccines-12-00844-f003:**
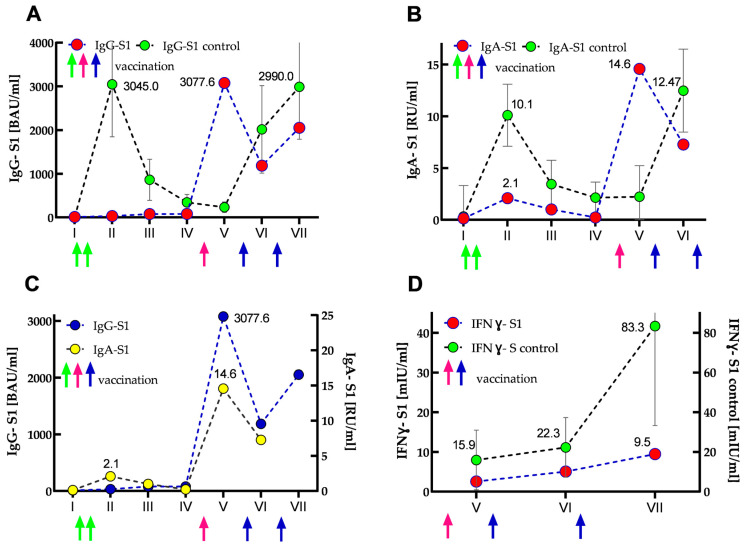
Kinetics of IgG-S1 and IgA-S1 antibodies (**A**–**C**), and IFNγ-S1 (**D**) for the SOTR case and control group (immunocompetent subjects) presented as median (and IQR for the control group). Circles and arrows indicate anti-S1 antibodies/IFNγ-S1 levels and vaccine dose administrations (green—primary vaccination, pink—SOTR’s additional dose; blue—booster doses of the control group), respectively. (**A**) IgG-S1 antibody kinetics for the case (marked in red) vs. control group (marked in green). (**B**) IgA-S1 antibody kinetics for the SOTR case (marked in red) vs. control group (marked in green). (**C**) IgG-S1 antibody kinetics (marked in blue) vs. IgA-S1 antibody kinetics (marked in yellow) for the SOTR case. (**D**) Kinetics of normalized data of IFNγ-S1 (marked in red) for the SOTR case vs. IFNγ-S1 kinetics (marked in green) for the control group.

**Table 1 vaccines-12-00844-t001:** Kinetics of IgG-S1 and IgA-S1 antibodies at checkpoints for the SOTR and control group.

	Variable	IgG-S1 (BAU/mL)							IgA-S1 (RU/mL)					
		I	II	III	IV	V	VI	VII	I	II	III	IV	V	VI
Control	N	88	88	88	87	84	83	74	88	88	88	87	84	83
	Median	11.90	3045	861	345	233	2015	2990	0.32	10.1	3.44	2.14	2.23	12.47
	IQR	9.40	2490	942	370	252	2056	3481	6.12	6.48	4.70	3.23	6.19	18.53
	Percentage	0.39	100	30.61	11.58	8.27	50.68	83.31	3.16	100	38.36	24.46	23.71	104.31
SOTR	Value Percentage	11.78 0.38	32.0 1.05	77.81 2.55	79.21 2.60	3077.6 101.07	1185.6 38.93	2052.9 67.41	0.13 1.28	2.09 20.69	1.0 9.90	0.22 2.17	14.57 144.25	7.27 71.98

Observations: Roman numerals (I–VII), subsequent serum collection checkpoints; bold values of IgG-S1 and IgA-S1, with the highest antibody levels obtained for the control group and case; IQR, inter-quartile range. Percentages were calculated based on the values obtained for the control group at the second checkpoint.

**Table 2 vaccines-12-00844-t002:** The levels of IFNγ-S1 and non-specific IFNγ at checkpoints for the SOTR case and control group.

Variable			IFNγ-S1 (mIU/mL)			Non-Specific IFNγ (mIU/mL)		
			V	VI	VII	V	VI	VII
Raw data	Control	N	78	78	76	78	78	76
		Median	1303.2	1590.4	4250.5	7283.2	7127.1	4927.9
		IQR	1362.4	1577.7	5002.1	8533.6	6209.6	4022.4
		Percentage	100	122.1	326.1	100	97.6	67.6
	SOTR	Value Percentage	15.8 1.2	39.7 3.1	48.9 3.75	619.6 8.5	786.6 10.8	516.9 7.1
Normalized data *	Control	N	78	78	76	-	-	-
		Median	15.9	22.3	83.3			
		IQR	33.0	29.7	159.6			
		Percentage	100	140.3	1003.7			
	SOTR	Value Percentage	2.6 16.4	5.1 32.1	9.5 59.7	-	-	-

Observations: *, data of specific reactivity of immune cells to the S1 virus protein were normalized to the maximum response of all immune cells to a non-specific activator (IFNγ-S1/IFNγ (non-specific activator) × 100 for individual subjects, and the table shows the median of the results for consecutive collections; Roman numerals (V–VII), subsequent serum collection checkpoints; IQR, inter-quartile range. Percentages were calculated based on the value obtained for the control group at the fifth checkpoint.

## Data Availability

All relevant data can be found within the manuscript.

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
