# Peer review of "Comparison of Post-Vaccination Response (Humoral and Cellular) to BNT162b2 in Clinical Cases, Kidney and Pancreas Transplant Recipient with Immunocompetent Subjects over Almost Two Years of Parallel Monitoring"

_vaccines, 2024, doi:10.3390/vaccines12080844_

Round 1
Reviewer 1 Report
Comments and Suggestions for Authors
Comparison of post-vaccination response (humoral and cellular) to BNT162b2 in kidney and pancreas transplant recipient with immunocompetent subjects over almost two years of parallel monitoring
Jaroslaw Walory 1,*, Iza Ksiazek 2, Karolina Wegrzynska 1 and Anna Baraniak 1,*
1. The work is based on the comparison of a group of 91 controls with an immunosuppressed patient after a pancreas transplant. I believe that the title should specify that it is considered a clinical case.
2. Tables
2.1 Table 1 You must separate the line in which Value appears from the percentage, they are together.
2.2 In Table 2, explain broadly what value is used to normalize.
3. At no time is a study done on the cells of the immune system that intervene in the immune response, only the levels of interferon are expressed, I think that staying only on this molecule is a bit left on the surface of the study.
4. There is a disproportion between the number of controls and the case study, it would not be necessary to put such a high number of controls and then compare it with a single case, this study would remain merely descriptive and a greater number of cases is required. A more complete study could be carried out with immunosuppressed patients.
5. I recommend using flow cytometry to obtain more information on the cell populations involved (B lymphocytes, memory B lymphocytes, T lymphocytes, memory T lymphocytes, etc.) as well as specific cellular subpopulations expressed in the presence of viruses (study of dextramers, see Immudex commercial company.)
Author Response
Comment 1: The work is based on the comparison of a group of 91 controls with an immunosuppressed patient after a pancreas transplant. I believe that the title should specify that it is considered a clinical case.
Response 1: Thank you for pointing this out. We agree with this comment and we have changed the manuscript title (line 3). The title now reads: “Comparison of post-vaccination response (humoral and cellular) to BNT162b2 of a clinical case of kidney and pancreas transplant recipient with immunocompetent subjects over almost two years of parallel monitoring” (lines: 2-5).
Comment 2: Tables
2.1 Table 1 You must separate the line in which Value appears from the percentage, they are together.
Response 2.1: Thank you for pointing this out. We have corrected Table 1.
2.2 In Table 2, explain broadly what value is used to normalize.
Response 2: Thank you for pointing this out. This has already been explained in the description to Table 2: “data of specific reactivity of immune cells to the S1 virus protein were normalized to the maximum response of all immune cells to a non-specific activator (IFNγ-S1 / IFNγ (non-specific activator)x100” (lines: 187-189). To the above sentence we have added: "for individual subjects, and the table shows the median of the results for consecutive collections" (lines: 189-190).
Comment 3: At no time is a study done on the cells of the immune system that intervene in the immune response, only the levels of interferon are expressed, I think that staying only on this molecule is a bit left on the surface of the study.
Response 3: We agree with this comment, but the investigation of immune response levels was conducted in our diagnostic laboratory in order to issue results to patients determining the levels of their humoral and cellular responses obtained after the baseline and booster dose(s). Therefore, the purpose of the study was not to analyse the cellular response in depth, but to determine and compare the levels of immune response determined for the transplant recipient with those obtained for immunocompetent individuals during nearly two years of parallel monitoring.
Comment 4: There is a disproportion between the number of controls and the case study, it would not be necessary to put such a high number of controls and then compare it with a single case, this study would remain merely descriptive and a greater number of cases is required. A more complete study could be carried out with immunosuppressed patients.
Response 4: We agree with your suggestion, however, as we mentioned earlier, the study of immune response levels was carried out in a diagnostic laboratory for individuals (employees of our institute) who self-referred to the study to obtain results for their immune responses to particular vaccine doses. In the study population, only one woman had organ recipient status. Had we had more such patients we would have presented the data obtained for them, and the manuscript would not have been in the form of a brief report but would have been submitted as a full article. The qualified number of subjects in the study is representative of the Polish adult population (30 million), assuming a confidence in the results obtained of α = 0.95 and a maximum result error of up to 10% (we have added this information, lines: 98-100).
Comment 5: I recommend using flow cytometry to obtain more information on the cell populations involved (B lymphocytes, memory B lymphocytes, T lymphocytes, memory T lymphocytes, etc.) as well as specific cellular subpopulations expressed in the presence of viruses (study of dextramers, see Immudex commercial company.)
Response 5: Thank you for this valuable comment, but as we mentioned earlier, the purpose of the study was not to explore the cellular response in depth. However, your recommendation is valid and we will try to implement it in our other studies. In this case, the investigation was conducted from January 2021 to October 2022, and the study material was patients' plasma, so blood morphotic elements were removed. All this makes it impossible for us to carry out the indicated tests.
Reviewer 2 Report
Comments and Suggestions for Authors
The authors performed a study on the comparison of post-vaccination response to BNT162b2 in kidney and pancreas transplant recipients! This study aimed to shed light on the humoral and cellular responses to the COVID-19 vaccine in immunocompromised individuals. It is helpful for COVID19 vaccine evaluation.
Here are some questions and suggestions
1) The immune response obtained by stimulating T cells with a non-specific mitogen is not sufficiently specific. The authors should mention this limitation in the discussion.
2) Why do all curves in Figure 3 lack error bars?
Author Response
Comment 1: The immune response obtained by stimulating T cells with a non-specific mitogen is not sufficiently specific. The authors should mention this limitation in the discussion.
Response 1: We agree with your opinion that the immune response achieved by stimulating T cells with a non-specific mitogen is not sufficiently specific, however, it is a good reference used by the test manufacturer. Therefore, to determine the specific cellular response to SARS-CoV-2, T cells were treated with the S1 SARS-CoV-2 antigen and the released IFN-γ-S1 was measured (we described this in lines 91-93). The immune response obtained by stimulating T cells with a non-specific mitogen was carried out to determine the total cellular non-specific response, which allowed to normalize the data obtained for the specific response (we described this in lines: 188-191; Table 2). As you suggested, we have added a Limitations section and included the restrictions of our study (line 249).
Comment 2: Why do all curves in Figure 3 lack error bars?
Response 2: Thank you for pointing this out. We have corrected Figure 3.
Reviewer 3 Report
Comments and Suggestions for Authors
This study aims to compare the post-vaccination immune response to BNT162b2 (Pfizer-BioNTech COVID-19 vaccine) in kidney and pancreas transplant recipients with immunocompetent individuals over an extended period of two years. This research topic is highly relevant given the increased risk of severe COVID-19 outcomes among solid organ transplant recipients (SOTRs) and the need to optimize vaccination strategies for this vulnerable population.
Majors,
!# Sample Size and Representativeness: The reviewer notes that the sample size and demographic information of the study participants are not provided. It is essential to ensure that the sample is adequately powered and representative of the target population to generalize the findings.
2# Study Design: This part provides an insufficiently clear introduction to the vaccination plan, which is prone to causing confusion and misunderstandings. A detailed elaboration should be made on the vaccination plan for the patients in this part and the significance and purpose of conducting this experimental design. In this study, the experimental design should achieve strict control of variables as much as possible.
3#Limitations and revision directions: The study should acknowledge its limitations, such as the potential for confounding factors (e.g., variations in immuno-suppressive therapies, comorbidities) that may impact the immune responses observed. Furthermore, exploring the effects of booster doses or alternative vaccines, would be valuable.
This study has the potential to provide important insights into the immune response to COVID-19 vaccination in kidney and pancreas transplant recipients. With thorough revisions to address the above-mentioned areas for improvement, the study could make a significant contribution to the field.
Comments on the Quality of English Language
Minor editing of English language required
Author Response
Comment 1: Sample Size and Representativeness: The reviewer notes that the sample size and demographic information of the study participants are not provided. It is essential to ensure that the sample is adequately powered and representative of the target population to generalize the findings.
Response 1: Thank you for pointing this out. We have added information on the representativeness of the group (lines: 98-100) and the main demographics of the study participants classified as the control group (lines: 100-102); details have been published previously (https://doi.org/10.3390/vaccines11101578).
Comment 2: Study Design: This part provides an insufficiently clear introduction to the vaccination plan, which is prone to causing confusion and misunderstandings. A detailed elaboration should be made on the vaccination plan for the patients in this part and the significance and purpose of conducting this experimental design. In this study, the experimental design should achieve strict control of variables as much as possible.
Response 2: Thank you for your valuable comment, however, our study was designed to compare the post-vaccination response to BNT162b2 in kidney and pancreas transplant recipients with immunocompetent individuals over nearly two years of simultaneous monitoring. However, at your suggestion, we have revised the manuscript to include compliance of COVID-19 vaccination in immunocompromised individuals, in accordance with the current recommendations of the European Centre for Disease Prevention and Control, ECDC (https://www.ecdc.europa.eu/sites/default/files/documents/Public-health-considerations-to-support-decisions-on-implementing-a-second-mRNA-COVID-19-vaccine-booster-dose.pdf). So, our solid organ transplant recipient case received only three doses of COVID-19 vaccination, which according to current ECDC recommendations is the primary vaccination (two doses) and one additional dose, and therefore did not receive any booster dose. Considering this, we have changed a little bit our conclusions.
Comment 3: Limitations and revision directions: The study should acknowledge its limitations, such as the potential for confounding factors (e.g., variations in immuno-suppressive therapies, comorbidities) that may impact the immune responses observed. Furthermore, exploring the effects of booster doses or alternative vaccines, would be valuable.
Response 3: Thank you for pointing this out. We have added a Limitations section. Our study included only one organ recipient and we defined this as its limitation (line 249). In addition, in the Results, Participant Characteristics section, we added values for the daily dose of immunosuppressive maintenance medications (lines: 106-107). We have previously discussed the effects of these medications on the post-vaccination immune response in the Discussion section. (lines 209-218). Regarding the patients in the control group, as we described previously, their comorbidities were not found to affect their post-vaccination humoral response (https://doi.org/10.3390/vaccines11101578).
Round 2
Reviewer 3 Report
Comments and Suggestions for Authors
All the comments have been addressed.
In the revised title, "clinical cases" may be more suitable than "a clinical case". And in the main body there are lots of "case" ought to be "cases"
Comments on the Quality of English LanguageMinor editing of English language required
Author Response
Comment 1: In the revised title, "clinical cases" may be more suitable than "a clinical case". And in the main body there are lots of "case" ought to be "cases"
Response 1: We have corrected the title with the reviewer's suggestion.
Comment 2: Minor editing of English language required.
Response 2: We have corrected minor language mistakes.